# A Novel Micro-Contact Stiffness Model for the Grinding Surfaces of Steel Materials Based on Cosine Curve-Shaped Asperities

**DOI:** 10.3390/ma12213561

**Published:** 2019-10-30

**Authors:** Qi An, Shuangfu Suo, Fuyan Lin, Jianwen Shi

**Affiliations:** 1School of Mechanical Electronic & Information Engineering, China University of Mining & Technology-Beijing, Beijing 10083, China; linfy@cumtb.edu.cn; 2Department of Mechanical Engineering, State Key Laboratory of Tribology, Tsinghua University, Beijing 100084, China; sfsuo@tsinghua.edu.cn (S.S.); shjw6222@mail.tsinghua.edu.cn (J.S.)

**Keywords:** steel material, grinding surface, simulated rough surface, microcontact stiffness model, asperity, elastic–plastic deformation

## Abstract

Contact stiffness is an important parameter for describing the contact behavior of rough surfaces. In this study, to more accurately describe the contact stiffness between grinding surfaces of steel materials, a novel microcontact stiffness model is proposed. In this model, the novel cosine curve-shaped asperity and the conventional Gauss distribution are used to develop a simulated rough surface. Based on this simulated rough surface, the analytical expression of the microcontact stiffness model is obtained using contact mechanics theory and statistical theory. Finally, an experimental study of the contact stiffness of rough surfaces was conducted on different steel materials of various levels of roughness. The comparison results reveal that the prediction results of the present model show the same trend as that of the experimental results; the contact stiffness increases with increasing contact pressure. Under the same contact pressure, the present model is closer to the experimental results than the already existing elastic–plastic contact (CEB) and finite-element microcontact stiffness (KE) models, whose hypothesis of a single asperity is hemispherical. In addition, under the same contact pressure, the contact stiffness of the same steel material decreases with increasing roughness, whereas the contact stiffness values of different steel materials under the same roughness show only small differences. The correctness and accuracy of the present model can be demonstrated by analyzing the measured asperity geometry of steel materials and experimental results.

## 1. Introduction

The surfaces of machined parts are not completely smooth. The contact stiffness of rough surfaces directly affects the connection performance between mechanical parts and has an important influence on the stability and reliability of mechanical systems [1,2]. Owing to the increased precision requirements for mechanical products, most important surfaces in mechanical systems are processed by grinding. Contact stiffness is an important parameter for describing the contact behavior of rough surfaces. If the contact stiffness between the grinding surfaces of steel materials can be described accurately, this accurate description will play a guiding role in solving practical engineering problems.

The contact stiffness of rough surfaces has always been an important topic in the field of tribology. Greenwood and Williamson [3] were the first to propose a random simulated rough surface based on the hypothesis of hemispherical asperities and statistical theory. Combining rough surface simulation and the Hertz contact theory, an analytical rough-interface contact model (GW model) was proposed in 1966. In 1987, Chang [4] put forward an elastic–plastic contact model (CEB model) for rough surfaces based on volume conservation of asperity control volume during plastic deformation. Since then, many scholars have conducted extensive research based on simulated rough surfaces of hemispherical asperity [5,6,7,8,9]. With the development of the finite-element technology, Kogut [10] used the finite-element method to analyze the contact problem between hemispherical asperities and a rigid plane. An empirical formula for contact stiffness was obtained, and the finite-element microcontact stiffness model (KE model) was established in 2003. Subsequently, much research work has been accomplished [11,12,13], revising and extending the already existing contact model. However, in the process of modifying the already existing model, the above authors focused on the analysis of the mechanical contact calculation and of the extended elastic–plastic deformation process for asperities under contact load. The adopted simulated rough surface in such analyses has been the same as the GW model, with no alternative model for simulated rough surfaces introduced.

Several scholars have since recognized the problems of simulated rough surfaces under different machining methods. Horng [14] proposed a hypothesis for asperities with a semiellipsoidal geometry in 1998 and then extended the CEB model to the general case of elliptical contact. The semiellipsoidal geometry hypothesis is an extension of the hemispherical geometry hypothesis [15]. Research work has addressed the asperity geometry problem of rough surfaces under different processing methods. However, owing to the limitations of the measurement technologies in the 20th century, the semiellipsoidal geometry hypothesis was not interrelated with the asperity geometry on the measured surface. Since then, some scholars have supplemented and expanded the microcontact stiffness model based on the semiellipsoidal asperity hypothesis [15,16,17,18].

The purpose of the present study is to explore the contact characteristics for the grinding surfaces of steel materials, as well as to establish a new method for analytical calculation of contact stiffness. A novel microcontact stiffness model which is more suitable for the grinding surfaces of steel materials is established, and simulations are carried out for the contact stiffness of grinding surfaces obtained with different levels of roughness, thereby providing a contact stiffness acquisition approach. Moreover, the correctness and accuracy of the present model are verified experimentally.

## 2. Establishment of a Simulated Rough Surface

### 2.1. Reconstruction of a Simulated Rough Surface

To make the established microcontact stiffness model more accurate, it is necessary to establish a simulated rough surface that is more consistent with physical specimen surfaces. A simulated rough surface is actually a hypothesis about the geometry of a single asperity and the height distribution of the asperities on the measured surface.

As mentioned in the Introduction, there are essentially two kinds of hypotheses about the geometry of a single asperity: hemispherical and semiellipsoidal. The hemispherical hypothesis is the most widely used at present. The hemispherical geometry hypothesis is, therefore, the research focus of this study. Since the vertical section shapes along the center point of various asperity geometries are the same, it was necessary to use the vertical section for analysis and processing. The acquisition process for the radius of curvature for the hemispherical geometry is described as follows. First, the peak point of a single wave peak on a rough surface is determined. According to the peak point, the left and right adjacent points are obtained. Based on these three points, the radius of curvature for the hemispherical geometry can be determined. By marking the number of wave peaks on the rough surface, the radius of curvature for each peak point is solved individually. Then, the radii of curvature are averaged. The final average radius of curvature is used as the initial value for the calculation of the contact stiffness.

To make the simulated rough surface more similar to real surfaces, the collected grinding surface topography data were analyzed and processed. Figure 1a shows the results of the ZYGO NexView noncontact microtopography measurement system (ZYGO Corporation, Middlefield, CT, USA) for a 304 stainless-steel grinding surface at a roughness (Sa) of 0.122 μm. Sa is the arithmetical mean height of the scale limited surface. The sampling area was 3 mm × 3 mm and the sampling number was 1024 × 1024. Figure 1b is a randomly selected vertical section image of the grinding surface. Based on the collected data for a single asperity randomly selected on the grinding surface, different geometric shapes were fit to the data points. The fitting results are shown in Figure 2a.

It should be noted that, for the grinding surface topography, the magnification ratios between the sampling length direction and the sampling height direction are displayed differently. This display difference does not affect the analysis process. It does, however, lead to a distortion characteristic on the displayed curve-fitting results. The hemispherical fitting results show a semielliptical shape, whereas the shape of the cosine curve remains unchanged.

The fitting result shows that when the hemispherical geometry is used to fit the data points of the asperity, the fitting curve can fit well around the peak point. However, a complete fit from the geometry peak point to the valley point could not be achieved. That is to say, the current fitting result of an actual asperity geometry had local characteristics that were not reflected by the overall characteristics.

In this investigation, a cosine curve is used to fit the collection of points of the whole profile of a single asperity on the grinding surface. Such a fitting result is shown in Figure 2a. It can be seen that a complete fitting from a valley point to a peak point to a valley point can be achieved using a cosine curve. By randomly selecting a single-asperity geometry for data fitting, it was found that the root mean square (RMS) error between the cosine function fitting curve and the data points can be controlled to less than 3%, whereas the error of the semicircular curve is above 12%. Therefore, in the present work, a semiperiodic cosine-curve revolving body was used in place of the already existing hemispherical and semiellipsoidal asperity geometry hypotheses. The vertical section of the semiperiodic cosine-curve revolving body can be found in Figure 2b.

The geometry of a single asperity was obtained through the above analysis, but the distribution of the asperities on the rough surface is analyzed below. As with the GW model [3], CEB model [4], and KE model [10], a Gaussian distribution was applied to the construction of a simulated rough surface in the present investigation. The expression of the Gauss distribution is shown in Equation (22), and the parameters of the Gauss distribution are calculated using methods from the literature.

### 2.2. Determination of the Parameters of the Simulated Rough Surface

From the above analysis, the geometry of a single asperity of the simulated rough surface was determined. The geometric dimensions can be defined as: (1)$$z\left(x\right)=h\cos\left(\frac{\pi x}{l}\right),\;-\frac{l}{2}<x<\frac{l}{2},$$
where l is the wavelength of an asperity, and h is the height of an asperity.

As can be seen from Equation (1), only the height and wavelength of the cosine curve are needed in this parameterization. The determination process of the two parameters proceeded as follows. The peak points and valley points of the grinding surface were marked. The average distance between adjacent peak points was obtained, and half the average value was taken as the wavelength of the cosine function. The average value of the difference in height between adjacent peaks and valleys was taken as the height of the cosine function. Thus, the geometric parameters of a single asperity were obtained.

## 3. Establishment of the Microcontact Stiffness Model for Grinding Surfaces

### 3.1. Axiomatic Hypotheses for Establishing the Microcontact Model

As in the CEB model and KE model, the following hypotheses were proposed for grinding surfaces: (1) the single-asperity geometry on the contact surface is a semiperiodic cosine-curve revolving body; (2) all asperities on grinding surfaces have congruent geometry, and the height distribution of the asperities obeys a Gaussian distribution; (3) the interaction between asperities can be safely ignored in the contact process; (4) the deformation of a macromatrix can be safely ignored in the contact process. In addition, the contact of two rough surfaces with respective RMS values of $\sigma_{1}$ and $\sigma_{2}$ can be safely reframed as a smooth rigid surface coming into contact with a rough surface with an RMS value of $\sigma =\sqrt{\sigma_{{}_{1}}^{2}+\sigma_{{}_{2}}^{2}}$ [19].

### 3.2. Establishment of the Microcontact Stiffness Model for a Single Asperity

Based on the above hypotheses, the contact stiffness of a single asperity under contact load can be deduced. The mechanical microcontact stiffness model of a single asperity is shown in Figure 3, where the dotted line represents the geometric shape of the asperity in the initial state and the solid line represents the geometric shape of the asperity under contact load. Moreover, α is the interference, r is the radius of curvature of the contact area between the asperity and the rigid plane, and d is separation based on asperity height. With increasing contact load, the asperity undergoes three stages of deformation: elastic deformation, elastic–plastic deformation, and fully plastic deformation. The deformation process is described in three stages below.

When the contact load is small and the asperity is in the stage of elastic deformation, the Hertz contact theory [20] is used. The relationship between the interference (α) and the average pressure ($f_{me}$) in the elastic contact area can be expressed as follows:(2)$$\alpha ={\left(\frac{3\pi f_{me}}{4E}\right)}^{2}r_{e}$$
where E is the equivalent Young’s modulus, $E={\left(\frac{1-\nu_{1}^{2}}{E_{1}}+\frac{1-\nu_{{}^{2}}^{2}}{E_{2}}\right)}^{-1}$, where $E_{1}$ and $E_{2}$ are Young’s moduli; $\nu_{1}$ and $\nu_{2}$ are Poisson’s ratios; and $r_{e}$ is the elastic contact radius of curvature of an asperity.

Hence, from Equation (2), the average contact pressure ($f_{me}$) during elastic contact can be obtained in the form:(3)$$f_{me}=\frac{4E}{3\pi }{\left(\frac{\alpha }{r_{e}}\right)}^{1/2}$$

The real contact area during the elastic contact stage is [20]:(4)$$A_{e}=\pi r_{e}\alpha$$

According to the theory of contact mechanics [21], the material will begin to yield when the average contact pressure reaches the yield strength (*σ*_*y*_). That is, when $f_{me}=\sigma_{y}$, the material will begin to plastically deform.

Substituting $\sigma_{y}$ into Equation (3), the critical interference for elastic deformation ($\alpha_{ec}$) can be obtained by substitution in (3) as follows: (5)$$\alpha_{ec}={\left(\frac{3\pi \sigma_{y}}{4E}\right)}^{2}r_{e}$$

Using Equations (3) and (5), the average contact pressure ($f_{me}$) in the elastic deformation stage can be written as:(6)$$f_{me}=\sigma_{y}{\left(\frac{\alpha }{\alpha_{ec}}\right)}^{1/2}$$

It can be seen from Equation (5) that, to obtain the value of the critical interference ($\alpha_{ec}$), it is also necessary to determine the contact radius of the asperity. From Equation (1), the contact radius of the contact area in the elastic deformation stage ($r_{e}$) can be obtained as follows: (7)$$r_{e}={\left|\frac{1}{\frac{d^{2}z\left(x\right)}{dx^{2}}}\right|}_{x=0}={\frac{l^{2}}{h\pi^{2}\cos\left(\frac{\pi x}{l}\right)}|}_{x=0}=\frac{l^{2}}{h\pi^{2}}$$

The elastic contact load is equal to the product of the average pressure ($f_{me}$) and the elastic contact area ($A_{e}$). Combined with Equations (4), (6), and (7), the relationship between the elastic contact load ($F_{e}$) and interference (α) can be obtained by:(8)$$F_{e}=\frac{\sigma_{y}l^{2}\alpha_{ec}^{-1/2}\alpha^{3/2}}{h\pi }$$

Contact stiffness is the derivative of the contact load with respect to interference. The expression for contact stiffness ($k_{e}$) in the elastic contact stage can be obtained as follows: (9)$$k_{e}=\frac{dF_{e}}{d\alpha }=\frac{3\sigma_{y}l^{2}}{2h\pi }{\left(\frac{\alpha }{\alpha_{ec}}\right)}^{1/2}$$

With increasing contact load, the asperity transitions from the elastic deformation stage to the elastic–plastic deformation stage. For the intermediate transition stage, the deformation situation is more complex and cannot be accurately described quantitatively. Transition processing is performed by constructing a template function [22]. Therefore, before analyzing the stage of elastic–plastic deformation, we first analyzed the stage of fully plastic deformation.

The relationship between the contact area and interference in the fully plastic deformation stage can be expressed as [23]:(10)$$A_{p}=2\pi \alpha r_{p}$$
where $r_{p}$ is the contact radius of the contact area in the fully plastic deformation stage.

Combining Figure 3 with Equation (1), the contact radius ($r_{p}$) of the contact area in the fully plastic deformation stage can be obtained as:(11)$$r_{p}=\frac{l}{\pi }\arccos\left(\frac{h-\alpha }{h}\right)$$

According to the theory of contact mechanics [21], when the average pressure is approximately equal to three times the yield strength, a fully yielded stage will manifest. Therefore, at such a time, the critical average pressure is $f_{mp}=3\sigma_{y}$.

For the critical point of fully plastic deformation, the relationship between the average pressure and the yield strength of the semiperiodic cosine-curve asperity geometry can be expressed as [21]:(12)$$\frac{f_{mp}}{\sigma_{y}}=\frac{2}{3}\left[1+\ln\left(\frac{1}{3}\cdot \frac{E}{\sigma_{y}}\cdot \left|\frac{dz\left(r_{pc}\right)}{dr_{pc}}\right|\right)\right]=3$$
where $r_{pc}$ is the contact radius of the contact area at the critical point in the fully plastic deformation stage and $\left|\frac{dz\left(r_{pc}\right)}{dr_{pc}}\right|$ is the absolute value of the slope at the $r_{pc}$ point.

From Equation (12), $r_{pc}$, the radius at the critical point in the fully plastic deformation, can be obtained. In combination with Equation (10), the critical interference for fully plastic deformation can be obtained as follows:(13)$$\alpha_{pc}=\frac{l}{2\pi }\arcsin\left[3e^{\frac{7}{2}}\frac{\sigma_{y}}{E}\frac{l}{\pi h}\right]$$

The plastic contact load is equal to the product of the average pressure ($f_{mp}$) and the plastic contact area ($A_{p}$). The relationship between the plastic contact load ($F_{p}$) and interference (α) in the fully plastic deformation can be obtained as follows:(14)$$F_{p}=6\pi \alpha r_{p}\sigma_{y}$$

From the relationship between the contact load and interference, the contact stiffness ($k_{p}$) in the fully plastic deformation can be deduced as follows:(15)$$k_{p}=\frac{dF_{p}}{d\alpha_{p}}=6\pi r_{p}\sigma_{y}$$

By determining the critical point for elastic–plastic deformation of the asperity, we can determine the stages during which the asperity undergoes elastic and fully plastic deformation. Transition processing is performed by constructing a template function, which can be written in the following form [22]:(16)$$f\left(\alpha \right)=-2{\left(\frac{\alpha -\alpha_{ec}}{\alpha_{pc}-\alpha_{ec}}\right)}^{3}+3{\left(\frac{\alpha -\alpha_{ec}}{\alpha_{pc}-\alpha_{ec}}\right)}^{2}$$

From the template function, the contact area of the elastic–plastic deformation stage ($A_{ep}$) can be expressed as:(17)$$A_{ep}=A_{e}+\left(A_{p}-A_{e}\right)f\left(\alpha \right)$$

Similarly, the average contact pressure in the elastic–plastic deformation stage ($f_{mep}$) is:(18)$$f_{mep}=f_{me}+\left(f_{mp}-f_{me}\right)f\left(\alpha \right)$$

As in the elastic deformation stage, the elastic–plastic contact load is equal to the product of the average pressure ($f_{mep}$) and the contact area ($A_{ep}$). Therefore, in combination with Equations (4), (6), (10), (12), (17), and (18), the expression for contact load ($F_{ep}$) in the elastic–plastic deformation stage can be obtained as follows:(19)$$F_{ep}=\left[\pi r_{e}\alpha +\left(2\pi r_{p}\alpha -\pi r_{e}\alpha \right)f\left(\alpha \right)\right]\times \left[\sigma_{y}\alpha_{ec}^{-1/2}\alpha^{1/2}+\sigma_{y}\left(3-\alpha_{ec}^{-1/2}\alpha^{1/2}\right)f\left(\alpha \right)\right]$$

The contact stiffness in the elastic–plastic deformation stage is: (20)$$k_{ep}=\frac{dF_{ep}}{d\alpha }$$

### 3.3. Establishment of the Microcontact Stiffness Model for Grinding Surfaces

From the analysis in the above section, we can obtain the analytical expressions relating the contact parameters to the contact load at different deformation stages for a single asperity. Based on a statistical analysis, the contact stiffness of the whole contact area can be obtained by integrating all asperities in the contact area. First, the number of asperities (n) actually involved in contact is defined as follows [4,24]: (21)$$n=\eta A_{n}\int_{d}^{\infty }\Phi \left(z\right)dz$$
where η is the areal density of asperities, $A_{n}$ is the nominal contact area, and Φ(z) is the distribution function of asperity heights.

As in the single-asperity deformation analysis, the contact analysis of the whole contact area will also be conducted in three different stages.

When the dimensionless interference $\alpha^{*}\leq \alpha_{{}_{ec}}^{*}$, an asperity is in the elastic deformation stage. Using the statistical analysis of the asperity number on the contact surface, the elastic contact load of two rough surfaces ($F_{{}_{e}}^{*}$) can be obtained using Equations (8) and (21) as follows:(22)$$F_{{}_{e}}^{*}=\frac{\eta A_{n}\sigma_{y}l^{2}\alpha_{ec}^{*-1/2}}{h\pi }\int_{d^{*}-y_{s}^{*}}^{d^{*}-y_{s}^{*}+\alpha_{{}_{ec}}^{*}}\alpha^{*3/2}\Phi^{*}\left(z^{*}\right)dz^{*}$$
All parameters in the equations are dimensionless, where $d^{*}=d/\sigma$, $\alpha^{*}=\alpha /\sigma$, $z^{*}=z/\sigma$, $y_{s}^{*}=y_{s}/\sigma$, $\alpha_{ac}^{*}=\alpha_{ac}/\sigma$, $\alpha_{pc}^{*}=\alpha_{pc}/\sigma$, and $\Phi^{*}\left(z^{*}\right)={\left(2\pi \right)}^{-1/2}\left(\frac{\sigma }{\sigma_{s}}\right)\exp\left[-0.5{\left(\frac{\sigma }{\sigma_{s}}\right)}^{2}z^{*2}\right]$. σ is the standard deviation of surface heights, and $\sigma_{s}$ is the standard deviation of asperity heights. $\frac{\sigma_{s}}{\sigma }={\left[1-\frac{3.717\times 10^{-4}}{{\left(\sigma R\eta \right)}^{2}}\right]}^{1/2}$, $\alpha^{*}=z^{*}-d^{*}+y_{s}^{*}$, and $d^{*}=e^{*}+y_{s}^{*}$, where $y_{s}$ is the difference between the mean of asperity heights and that of surface heights, $z^{*}=z/\sigma$, $y_{s}^{*}=y_{s}/\sigma$, $\alpha_{ac}^{*}=\alpha_{ac}/\sigma$, $\alpha_{pc}^{*}=\alpha_{pc}/\sigma$, and $\Phi^{*}\left(z^{*}\right)={\left(2\pi \right)}^{-1/2}\left(\frac{\sigma }{\sigma_{s}}\right)\exp\left[-0.5{\left(\frac{\sigma }{\sigma_{s}}\right)}^{2}z^{*2}\right]$.

Specific parameters are calculated using methods from the literature [4].

The contact stiffness in the elastic contact stage can be obtained from Equations (9) and (21) as follows:(23)$$K_{{}_{e}}^{*}=\frac{3\eta A_{n}l^{2}\sigma_{y}\alpha_{ec}^{*-1/2}}{2h\pi }\int_{d^{*}-y_{s}^{*}}^{d^{*}-y_{s}^{*}+\alpha_{{}_{ec}}^{*}}\alpha^{*1/2}\Phi^{*}\left(z^{*}\right)dz^{*}$$

When the dimensionless interference $\alpha_{{}_{ec}}^{*}\leq \alpha^{*}\leq \alpha_{{}_{pc}}^{*}$, an asperity is in the elastic–plastic deformation stage. Combining Equations (19) and (21), the elastic–plastic contact load of two rough surfaces ($F_{ep}^{*}$) can be written as:(24)$$F_{{}_{ep}}^{*}=\eta A_{n}\pi \sigma_{y}\int_{d^{*}-y_{s}^{*}+\alpha_{{}_{ec}}^{*}}^{d^{*}-y_{s}^{*}+\alpha_{{}_{pc}}^{*}}\left[\begin{array}{l} \alpha_{ec}^{*-1/2}\alpha^{*}{}^{1/2}+\\ \left(3-\alpha_{ec}^{*-1/2}\alpha^{*}{}^{1/2}\right)f^{*}\left(\alpha^{*}\right) \end{array}\right]\times \left[r_{e}+\left(2r_{p}-r_{e}\right)f^{*}\left(\alpha^{*}\right)\right]\alpha^{*}\Phi^{*}\left(z^{*}\right)dz^{*}$$

The contact stiffness ($K_{{}_{ep}}^{*}$) in the elastic–plastic deformation stage is:(25)$$K_{{}_{ep}}^{*}=\frac{dF_{{}_{ep}}^{*}}{d\alpha^{*}}$$

When the dimensionless interference $\alpha^{*}\geq \alpha_{{}_{pc}}^{*}$, the asperity is in the fully plastic deformation stage. Combining Equations (19) and (21), the plastic contact load of two rough surfaces ($F_{p}^{*}$) can be expressed as:(26)$$F_{{}_{p}}^{*}=6\pi r_{p}\sigma_{y}\int_{d^{*}-y_{s}^{*}+\alpha_{{}_{pc}}^{*}}^{\infty }\alpha^{*}\Phi^{*}\left(z^{*}\right)dz^{*}$$

The contact stiffness ($K_{{}_{p}}^{*}$) in the fully plastic deformation stage is: (27)$$K_{{}_{p}}^{*}=6\pi r_{p}\sigma_{y}\eta A_{n}\int_{d^{*}-y_{s}^{*}+\alpha_{{}_{pc}}^{*}}^{\infty }\Phi^{*}\left(z^{*}\right)dz^{*}$$

The above analytical expressions govern the contact parameters for the whole contact surface. The contact load and stiffness of the whole contact surface are summed over all the asperities on the rough surface in different deformation stages. Therefore, the total contact load ($F^{*}$) on the contact surface at any given time can be expressed as:(28)$$F^{*}=F_{{}_{e}}^{*}+F_{{}_{ep}}^{*}+F_{{}_{p}}^{*}$$

The total contact stiffness ($K_{w}^{*}$) on the contact surface can be expressed as:(29)$$K_{w}^{*}=K_{{}_{e}}^{*}+K_{{}_{ep}}^{*}+K_{{}_{p}}^{*}$$

To facilitate comparison, what is compared and analyzed in the analysis below is contact stiffness under unit area contact pressure. That is, the contact pressure is divided by the nominal contact area, as shown in the following equation:(30)$$F_{w}^{*}=F^{*}/A_{n}$$

## 4. Experimental Verification

### 4.1. Preparation of Specimens

The experimental research took three different steel materials as the research objects: 45# steel, 40Cr steel, and 304 stainless steel. First, the specimens were machined to achieve the required size and shape. Common grinding wheels of various granularities (46#, 80#, and 120#) were selected to grind the specimen surfaces. Then, grinding surfaces with different levels of roughness were obtained. For each material, each of two specimens had the same roughness. Each sample was cleaned ultrasonically and dried. Owing to space limitations, we used 304 stainless steel as an example. Figure 4 shows a picture of 304 stainless-steel specimens.

After the completion of specimen preparation, the next step was to obtain the relevant parameters, including material parameters (Young’s modulus E, Poisson’s ratio ν, yield strength $\sigma_{y}$, and hardness H) as well as the statistical parameters of the measured grinding surface (average radius of curvature R of asperities, average wavelength l of asperities, average height h of asperities, standard deviation σ of surface heights, and areal density η of asperities).

The acquisition of material parameters was based on conventional mechanical experiments. Table 1 shows the material parameters of the three steel materials. However, the statistical parameters of the measured grinding surface need to be processed from the topography data according to the previously described method. First, the three-dimensional topography of the specimens’ surfaces was assessed. The average values of the grinding surface roughness obtained by the different granularity grinding wheels (120#, 80#, and 46#) were Sa 0.122 μm, Sa 0.345 μm, and Sa 0.672 μm, respectively. The grinding surface roughness of the same grinding wheel is basically the same (the RMS error can be controlled within 0.5%). Therefore, the grinding surface roughness levels obtained by different granularity grinding wheels were directly expressed by the respective average values. By processing the collected topography data, the statistical parameters can be obtained and are shown in Table 2.

### 4.2. Test Rig

Equations (9), (15), and (20) show that the essence of contact stiffness is the ratio of the change in contact pressure to the change in interference. A rough surface contact stiffness test rig was built at Tsinghua University. A schematic diagram and physical drawings of the contact stiffness test rig are shown in Figure 5a,b, respectively. 

The whole test rig was vertically structured. All of the loading test units were connected in series in the support structure. A rubber mat was used for vibration isolation at the lower end of the support structure. The unit was loaded by a hydraulic jack with a target load of 6000 N. A KAP-TC pressure sensor (Kewill GmbH, Hamburg, Germany) was used to measure the contact load of the test surface. Three KD2306-1S eddy current displacement sensors (Kaman, Bloomfield, CT, USA) were installed on the upper specimens and uniformly arranged to measure the normal interference of the contact surface under contact load. The main parameters of experimental test are shown in Table 3.

Finally, through data acquisition, the variation curve of the contact load and the interference could be obtained. The curve describing the relationship between contact stiffness and contact load can be obtained by data processing. Each group of experiments was carried out using two specimens of the same material and the same roughness. Meanwhile, to ensure the reliability and repeatability of the data, three groups of samples were used for repeated experiments for each group of tests, and the average values were taken as the final data.

It is important to note that there is a distance between the sensor position and the test point of the contact surface. Therefore, this component of the elastic deformation of the matrix was subtracted from the test data of the sensor during the calculation of the experimental results.

## 5. Results and Discussion

In addition to the experimental results, the prediction results of the CEB and KE models were used for comparative analysis. The analytical expressions of the CEB [4] and KE [10] models are detailed in the literature. The initial values for the numerical simulation of the three models are all shown in Table 1 and Table 2.

Figure 6 shows the relationship between the contact stiffness ($K_{w}^{*}$) and the contact pressure ($F_{w}^{*}$) for different materials at different levels of roughness. The letters “a,” “b,” and “c” correspond to three different steel materials: 40Cr steel, 45# steel, and 304 stainless steel, respectively. The numbers “1,” “2,” and “3” correspond to three different levels of roughness: Sa 0.122 μm, Sa 0.345 μm, and Sa 0.672 μm, respectively.

Figure 6 shows that, for the relationship between the experimental results and the numerical simulation results, all the graphics are similar. Therefore, Figure 6a1 is taken as an example for a detailed analysis.

Figure 6a1 shows the relationship between the contact stiffness and contact pressure of 40Cr steel at a roughness of Sa 0.112 μm. As a whole, the predictions of the three models showed the same trend as the experimental results. The contact stiffness increases with increasing contact pressure, but there are differences between the values. Under the same contact pressure, the present model was shown to be closer to the experimental results compared to the CEB and KE models, whose asperity geometry is hemispherical. Locally, differences in contact stiffness differ across contact pressure ranges.

Within the range of contact pressure $F_{w}^{*}\leq 10$ MPa, the simulation results of the present model were basically consistent with those of other models. For contact pressure $F_{w}^{*}=10$ MPa, the contact stiffness values of the experimental results, present model, CEB model, and KE model were 17.32, 16.83, 16.17, and 16.49 MPa/μm, respectively. These results were not unexpected. At a low contact pressure, an asperity is in the elastic deformation stage. According to the asperity geometry fitting results Figure 2a, the radii of curvature of the asperity geometries with a semicircle and cosine curve were almost the same. Hence, in this stage, the values for contact stiffness ($K_{w}^{*}$) obtained by the three models were almost identical.

Within the range of contact pressure 10 MPa ≤$F_{w}^{*}$≤ 70 MPa, under the same contact pressure, the values for contact stiffness of the present model were greater than for the CEB and KE models. With increasing contact pressure, the differences in contact stiffness among the three models increase gradually. At contact pressure $F_{w}^{*}=$ 70 MPa, the contact stiffness reached its maximum in each of the three models, and the differences among the three models reached their maxima as well. At this point, the contact stiffness values were 103.71, 102.62, 83.02, and 97.73 MPa/μm for the experimental results, the present model, the CEB model, and the KE model, respectively. Similarly, an analysis was carried out in conjunction with Figure 2a. With increasing contact pressure, the contact of an asperity on the grinding surface undergoes a transition from the elastic deformation stage to the elastic–plastic deformation stage. The radius of curvature of the hemispherical asperity hence becomes smaller than under the cosine model, so the difference between the radii of curvature gradually increases. Therefore, under the same contact pressure, more asperities in the CEB and KE models will enter the plastic deformation stage, compared to the present model, resulting in the reduction of the contact stiffness values for those other models. In addition, the difference between the contact stiffness values gradually increases as the difference between the radii of curvature increases. The comparison results shown by other images in Figure 6 are similar to those in Figure 6a1, so they will not be repeated here.

For the relationship between the contact stiffness and contact pressure under different levels of roughness of the same material, 40Cr steel is taken as an example for analysis, since other materials follow the same pattern. Compared with Figure 6a1–a3, it can be seen that, under the same contact pressure, the contact stiffness value decreases with increasing roughness Sa. When the contact pressure $F_{w}^{*}$ reached 70 MPa, the contact stiffness values for Sa 0.122 μm, Sa 0.345 μm, and Sa 0.672 μm in the present model were 102.62, 84.71, and 60.02 MPa/μm, respectively. Meanwhile, the respective contact stiffness values obtained from the experimental results were 103.71, 85.82, and 62.10 MPa/μm. These results are in good agreement with the experimental results obtained by Fei Du [25] and Huifang Xiao [26] for measuring the contact stiffnesses of rough interfaces using the ultrasonic method.

According to the statistical parameters of different grinding specimens in Table 2, the size of asperities on rough surfaces increases gradually with increasing roughness Sa, whereas the areal density of asperities on rough surfaces decreases. From Equations (22), (25), and (27), the contact stiffness is always proportional to the areal density of asperities in each contact deformation stage. Meanwhile, the influence of other parameters on the contact stiffness produces different patterns in different deformation stages. The areal density of asperities seems to play a dominant role in governing the contact stiffness, eventually leading to a decrease in the contact stiffness values with increasing roughness.

For the relationship between the contact stiffness and contact pressure under the same roughness of different materials, Sa 0.122 μm is taken as an example for analysis, since other levels of roughness also follow the same pattern. Comparing with Figure 6a1,b1,c1, it can be seen that, under the same contact pressure, the contact stiffness values of different materials made only small differences. When the contact pressure ($F_{w}^{*}$) reached 70 MPa, the contact stiffness values of 40Cr steel, 45# steel, and 304 stainless steel in the present model were 102.62, 105.38, and 103.12 MPa/μm, respectively. Meanwhile, the corresponding contact stiffness values obtained from the experimental results were 103.71, 108.47, and 107.22 MPa/μm. From Equations (22), (25), and (27), it can be seen that, at the same roughness, the contact stiffnesses of different materials depended on the material parameters. Table 1 shows that, for different steel materials, the material parameters make a little difference, leading to only a small difference in the contact stiffness values at the same roughness.

## 6. Conclusions

This study proposed a novel micro-contact stiffness model for the grinding surfaces of steel materials based on cosine curve-shaped asperities. The following conclusions are drawn: 

(1) According to the measured grinding surface topography of steel materials, a novel simulated rough surface was proposed. Based on the simulated rough surface, an analytical expression of the novel microcontact stiffness model based on cosine curve-shaped asperities was obtained.

(2) The contact stiffness of different steel materials and specimens under different levels of roughness was obtained by using numerical and experimental studies. The comparison results show that prediction results in the presented model have the same trend as the experimental results, the value of contact stiffness increases with the increase of contact pressure. Under the same contact pressure, the present model in this paper is closer to the experimental results than the CEB and KE models in which the hypothesis of the single asperity is hemispherical. According to the analysis of measured asperity geometry on the grinding surface and the degree of conformity with the experimental results, the correctness and accuracy of the novel micro-contact stiffness model presented in this paper can be demonstrated.

(3) The presented model can describe the contact stiffness between grinding surfaces of steel materials more accurately, which provides guidance for the mechanical structure design and mechanical system stability analysis. However, further verification is needed to apply the model more widely (e.g., to rough surfaces formed by other materials or other processing methods).

## Figures and Tables

**Figure 1 materials-12-03561-f001:**
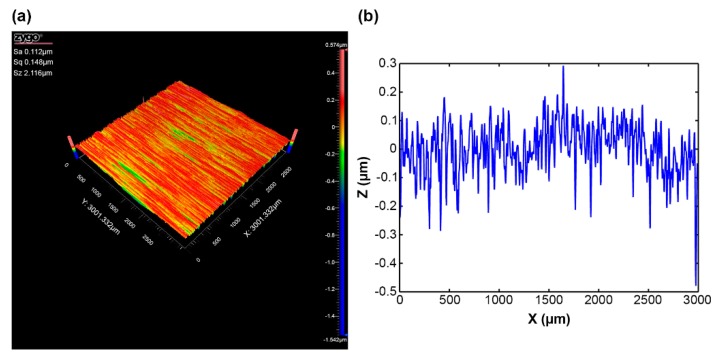
(**a**) Three-dimensional image of a 304 stainless-steel grinding surface at Sa 0.122 μm. (**b**) Vertical section image of a 304 stainless-steel grinding surface at Sa 0.122 μm.

**Figure 2 materials-12-03561-f002:**
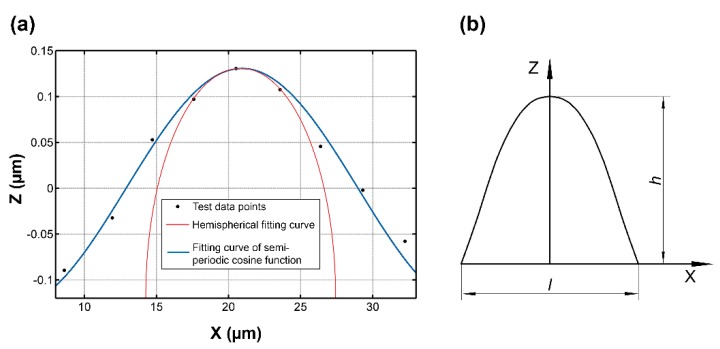
(**a**) Fitting results of measured data points using different fitting methods. (**b**) Vertical section image of a single asperity.

**Figure 3 materials-12-03561-f003:**
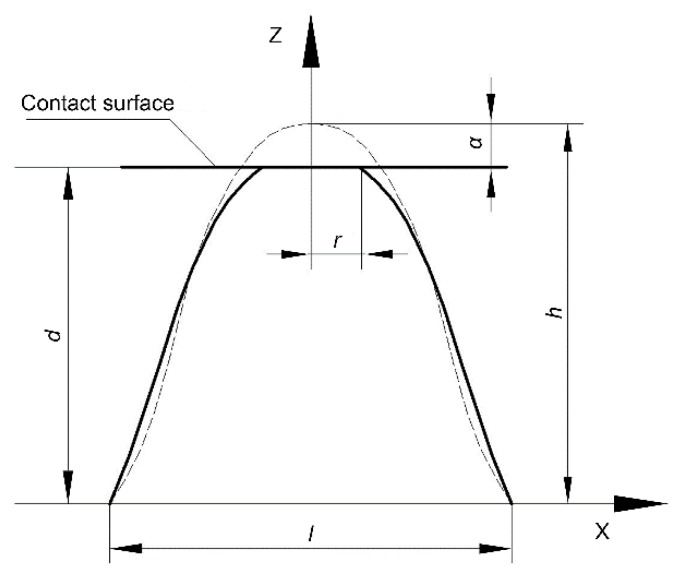
Microcontact stiffness model of a single asperity.

**Figure 4 materials-12-03561-f004:**
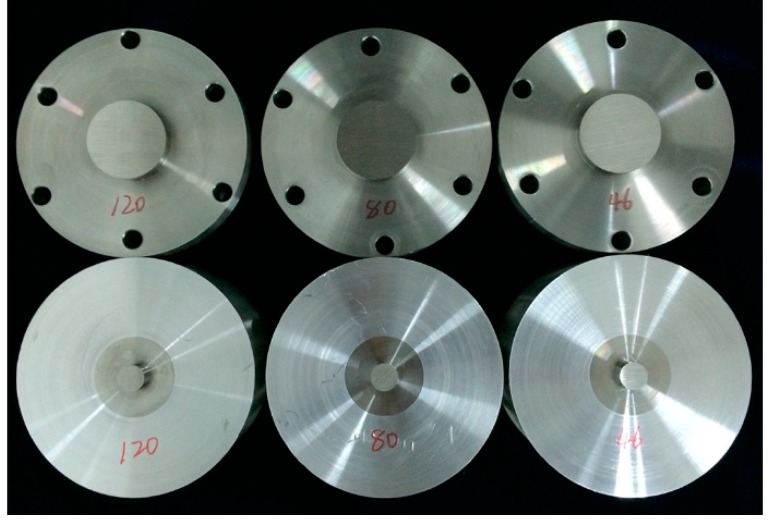
Specimens of 304 stainless steel.

**Figure 5 materials-12-03561-f005:**
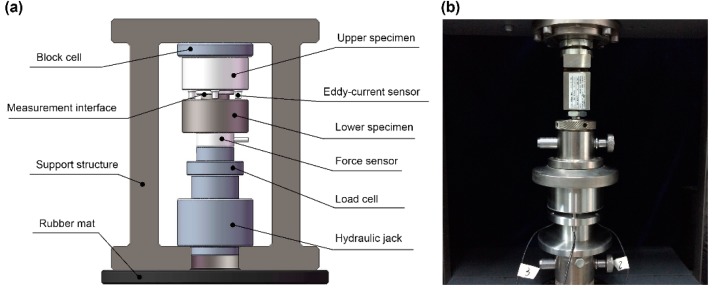
(**a**) A schematic diagram of the contact stiffness test rig. (**b**) Physical drawings of the contact stiffness test rig.

**Figure 6 materials-12-03561-f006:**
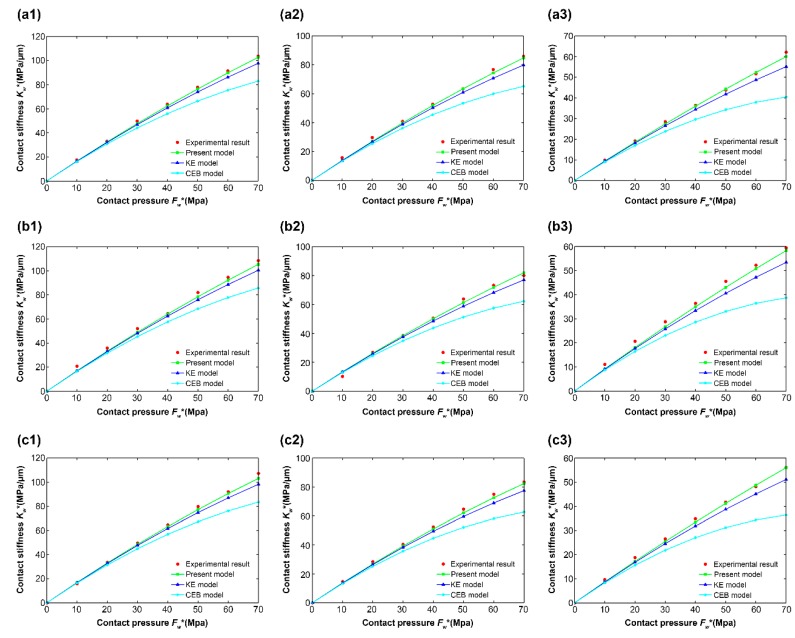
Comparison results of 45# steel for (**a1**) Sa 0.122 μm, (**a2**) Sa 0.345 μm, and (**a3**) Sa 0.672 μm. Comparison results of 40Cr steel for (**b1**) Sa 0.122 μm, (**b2**) Sa 0.345 μm, and (**b3**) Sa 0.672 μm. Comparison results of 304 stainless steel for (**c1**) Sa 0.122 μm, (**c2**) Sa 0.345 μm, and (**c3**) Sa 0.672 μm.

**Table 1 materials-12-03561-t001:** Material parameters of the three steel materials.

Material Type	*E* (GPa)	*υ*	*H* (MPa)	σ_y_ (MPa)
45#	210	0.269	1970	355
40Cr	211	0.290	2070	785
304	195	0.247	1870	205

**Table 2 materials-12-03561-t002:** Statistical parameters of the grinding surfaces for specimens.

Material Type	Wheel Granularity	Specimen Number	*R* (μm)	*l* (μm)	*h* (μm)	*σ* (μm)	*η* (mm^2^)
304	46	1	23.325	34.483	1.402	0.572	841
		2	23.319	34.379	1.396	0.567	846
	80	1	19.416	32.865	0.706	0.324	926
		2	19.425	32.971	0.712	0.326	920
	120	1	13.179	25.342	0.223	0.146	1557
		2	13.183	25.625	0.227	0.148	1523
45#	46	1	23.317	34.387	1.398	0.568	864
		2	23.323	34.416	1.413	0.569	861
	80	1	19.409	32.949	0.707	0.312	930
		2	19.414	32.953	0.714	0.319	926
	120	1	13.177	25.424	0.231	0.151	1506
		2	13.175	25.399	0.230	0.149	1500
40Cr	46	1	23.296	34.137	1.385	0.528	858
		2	23.302	34.345	1.379	0.532	847
	80	1	19.403	32.899	0.682	0.316	924
		2	19.415	32.941	0.699	0.322	922
	120	1	13.185	25.457	0.231	0.151	1543
		2	13.184	25.688	0.229	0.148	1515

**Table 3 materials-12-03561-t003:** The main parameters of experimental test.

Parameters		Value/Range
Loading rate		30 N/s
Loading range		0–5700 N
Sample frequency		10 kHz
Sampling resolution		16
KAP-TC pressure sensor	Output voltage range	±5 V
	Measuring range	0–6000 N
	Sensor sensitivity	2 mV/V
KD2306-1S eddy current displacement sensors	Output voltage range	0–10 V
	Measuring range	0–1 mm
	Sensor sensitivity	1000 mV/mm

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
