# Peer review of "A Novel Micro-Contact Stiffness Model for the Grinding Surfaces of Steel Materials Based on Cosine Curve-Shaped Asperities"

_materials, 2019, doi:10.3390/ma12213561_

Round 1

Reviewer 1 Report

This paper iinvestigates on the contact stiffness including elastic, elastic-plastic and fully plastic stage. It provides a more realistic model compared to the existing KE and CEB model.

  To show clearly the definition of Surface roughness  'Sa' that might be one of 'Rmax(peak-to-peak)', 'Rz(average 10 points)', 'Ra(waviness of middle line)'.Depending on the definition, the mechanical behavior would be totally different. 2. line 309-310, Roughness data should be listed corresponding to the grinding wheel numbers. 3. line 218-219, Could you check 'the elastic contact area(Ap)' to be 'the plastic contact area(Ap)'

Author Response

Response to Reviewer 1 Comments

The article has been revised one by one for the opinions of reviewer 2,The specific changes are as follows:

Point 1: To show clearly the definition of Surface roughness  'Sa' that might be one of 'Rmax(peak-to-peak)', 'Rz(average 10 points)', 'Ra(waviness of middle line)'.Depending on the definition, the mechanical behavior would be totally different.

 Response 1:

The modifications in section 2.1.

 is the arithmetical mean height of the scale limited surface.

(line 91-92)

Point 2: line 309-310, Roughness data should be listed corresponding to the grinding wheel numbers.

Response 2:

The modifications in section 4.1.

The average values of the grinding surface roughness obtained by the different granularity grinding wheels (120#, 80#, and 46#) were  0.122 μm,  0.345 μm, and  0.672 μm, respectively.

(line 320-322)

Point 3: line 218-219, Could you check 'the elastic contact area(Ap)' to be 'the plastic contact area(Ap)'.

Response 3:

The modifications in section 3.2.

the plastic contact area

(line 228)

Reviewer 2 Report

Reviewed article is very interesting and write at high scientific level. Presentation method is excellent and in accordance with generally accepted standards in that area. Figures, tables as well as terminology are clear and precise. Described method was correctly verified and compared with standard approach to this problem. Analysis are detailed and well described, scientific arguments were used to define the potential of presented method. Below are listed some comments that should be taken into consideration by the Authors to improve reviewed text:

at the end of the Introduction section authors should provide clear the aim and the novelty of the study on the basis of conclusions from state-of-the-art, not summary of presented work as it is in current form, text of scientific rapports should be write impersonal, I suggest to provide all main parameters and conditions of experimental test in a form of table, presented study widely covers defined scientific problem and with experimental investigations provides proper background for given conclusions, however deeper scientific consideration of obtained results referred to the basic phenomena should be given, I suggest to provide the main conclusions as numbered sentences and refer to specific values (results of analysis) as well as basic phenomena that cause described results, I suggest also to give wider description of potential use of presented findings in scientific research as well as in industrial practice.

Author Response

Response to Reviewer 2 Comments

The article has been revised one by one for the opinions of reviewer 1,The specific changes are as follows:

Point 1: At the end of the Introduction section authors should provide clear the aim and the novelty of the study on the basis of conclusions from state-of-the-art, not summary of presented work as it is in current form.

 Response 1:

The modifications in introduction section.

The purpose of the present study is to explore the contact characteristics for the grinding surfaces of steel materials, as well as to establish a new method for analytical calculation of contact stiffness. A novel microcontact stiffness model which is more suitable for the grinding surfaces of steel materials is established, and simulations are carried out for obtained contact stiffness of grinding surfaces with different levels of roughness, thereby providing a contact stiffness acquisition approach. Moreover, the correctness and accuracy of the present model are verified experimentally.

(line 64-69)

Point 2: Text of scientific rapports should be write impersonal, I suggest to provide all main parameters and conditions of experimental test in a form of table.

Response 2:

The modifications in section 4.2.

The main parameters of experimental test are shown in Table 3.

Table 3. The main parameters of experimental test

Parameters

Value/Range

Loading rate

30 N/s

Loading range

0-5700 N

Sample frequency

10 kHz

Sampling resolution

16

KAP-TC pressure sensor

Output voltage range

±5 V

Measuring range

0-6000 N

Sensor sensitivity

2 mV/V

KD2306-1S eddy current displacement sensors

Output voltage range

0-10 V

Measuring range

0-1 mm

Sensor sensitivity

1000 mV/mm

(line 351)

Point 3: Presented study widely covers defined scientific problem and with experimental investigations provides proper background for given conclusions, however deeper scientific consideration of obtained results referred to the basic phenomena should be given, I suggest to provide the main conclusions as numbered sentences and refer to specific values (results of analysis) as well as basic phenomena that cause described results, I suggest also to give wider description of potential use of presented findings in scientific research as well as in industrial practice.

Response 3:

The modifications in Conclusions section.

This study proposed a novel micro-contact stiffness model for the grinding surfaces of steel materials based on cosine-curve-shaped asperities. The following conclusions are drawn.

(1) According to the measured grinding surface topography of steel materials, a novel simulated rough surface was proposed. Based on the simulated rough surface, an analytical expression of the novel microcontact stiffness model based on cosine-curve-shaped asperities was obtained.

(2) The contact stiffness of different steel materials and specimens under different levels of roughness was obtained by using numerical and experimental studies. The comparison results show that prediction results in the presented model have the same trend as the experimental results, the value of contact stiffness increases with the increase of contact pressure. Under the same contact pressure, the present model in this paper is closer to the experimental results than the CEB model and KE model which the hypothesis of the single asperity is hemispherical. According to the analysis of measured asperity geometry on the grinding surface and the degree of conformity with the experimental results, the correctness and accuracy of the novel micro-contact stiffness model presented in this paper can be demonstrated.

(3) The presented model can describe the contact stiffness between grinding surfaces of steel materials more accurately, which provides guidance for the mechanical structure design and mechanical system stability analysis. However, further verification is needed to apply the model more widely (e.g., to rough surfaces formed by other materials or other processing methods).

(line 442-459)
